

# Insomnia partially mediates the relationship between pathological personality traits and depression: a case-control study

Fenglan Chen[1], Xiujin Lin[2], Yuli Pan[1], Xuan Zeng[3], Shengjie Zhang[1], Hong Hu[1], Miaoyu Yu[4] and Junduan Wu[1]

[1] Department of Psychology, School of Public Health, Guangxi Medical University, Nanning, Guangxi, China
[2] Department of Maternal and Child Health, School of Public Health, Sun Yat-Sen University, Guangzhou, Guangdong, China
[3] Department of Child Healthcare, Liuzhou Maternity and Child Healthcare Hospital, Liuzhou, Guangxi, China
[4] Department of Mental Health, The Second Affiliated Hospital of Guangxi Medical University, Nanning, Guangxi, China

Corresponding authors
Miaoyu Yu, yuliuyu01@sina.com
Junduan Wu,
wujunduan@gxmu.edu.cn

## ABSTRACT

**Background and Objective:** Personality disorders are frequently associated with insomnia and depression, but little is known about the inter-relationships among these variables. Therefore, this study examined these inter-relationships and the possible mediating effect of insomnia on the association between specific personality pathologies and depression severity.

**Methods:** There were 138 study participants, including 69 individuals with depression and 69 healthy controls. The main variables were measured by the Hamilton Depression Rating Scale-24 (HAMD-24), Athens Sleep Insomnia Scale (AIS), and the Personality Diagnostic Questionnaire (PDQ-4+). Multivariate linear regression and mediation analysis were conducted.

**Results:** With the exception of the antisocial personality score, all the PDQ-4+ scores and AIS scores were significantly higher in the depression group than in the healthy control group ($p < 0.001$). In the total sample, all personality pathology scores ($p < 0.001$), except the antisocial personality score, had significant positive correlations with the AIS scores and HAMD-24 scores, and the AIS scores and HAMD-24 scores were positively correlated ($r = 0.620$, $p < 0.001$). Regression analysis revealed that borderline personality, passive-aggressive personality, and insomnia positively predicted the severity of depression, after adjusting for sociodemographic covariates, and that insomnia partially mediated the associations of borderline personality and passive-aggressive personality with depression severity.

**Conclusions:** Borderline personality, passive-aggressive personality, and insomnia tend to increase the severity of depression, and the effect of borderline and passive-aggressive personality on depression severity may be partially mediated by insomnia. This is the first study to report these findings in a Chinese sample, and they may help researchers to understand the pathways from specific personality pathologies to the psychopathology of depression better, which should be useful for

designing interventions to relieve depression severity, as the impact of specific personality pathology and insomnia should be considered.

# INTRODUCTION

Depression is a common mental disorder, with a lifetime prevalence of depression being 3–17% in the global population (*Malhi & Mann, 2018*), and the prevalence of depression in China being 3.02% (*Smith, 2014*). However, the prevalence of depression in China varies among different study populations. For example, the point prevalence of major depression among Chinese children and adolescents is 1.3% (*Xu et al., 2018*), and 2.7% in older adults (*Wang et al., 2018*). Depression not only affects health outcomes, such as disability (*Eurviriyanukul et al., 2016*) and suicide behaviors (*Jung et al., 2018*), but also imposes a high economic burden (*Hsieh & Qin, 2018*), accounting for approximately 10.3% of the overall burden of diseases, worldwide (*Smith, 2014*). Therefore, it is necessary to explore and understand the risk factors for depression.

Personality disorders (PDs) are defined as patterns of inner experience and behavior that obviously deviate from the expectations of an individual's culture, have an onset in early life, and may lead to distress and impairment in individuals in the future (*Esbec & Echeburua, 2011*). According to the *DSM-IV* Criteria, PDs include cluster A PDs (paranoid, schizoid, and schizotypal PDs), cluster B PDs (histrionic, narcissistic, borderline, and antisocial PDs), cluster C PDs (avoidant, dependent, and obsessive PDs), and passive-aggressive and depressive PDs (see the Appendix of *DSM-IV*). A survey of 13 countries showed that the estimated prevalence of having any personality disorder was 6.1%, and the prevalence in China was 4.1% (*Huang et al., 2009*). Personality disorders, as a group, are among the most frequent disorders treated by psychiatrists (*Zimmerman, Rothschild & Chelminski, 2005*), which are associated with long-term mental illness and social consequences (*Samuels, 2011*). Currently, there is considerable support in the research literature for an association between depression and pathological personality; the presence of a pathological personality predicts the occurrence of later depression, and the severity of personality disorders increases the risk of major depressive disorder (*Moran et al., 2016*). For instance, during a 6-year follow-up study, 85% of persons with personality disorders had episodes of major depressive disorder, and 85% of those with lifetime major depressive disorder had recurrences (*Gunderson et al., 2008*). Another study found that personality disorders are comorbid with major depressive disorder, with 42.36% patients with major depressive disorder meeting at least one criterion for a diagnosis of a personality disorder (*Zheng et al., 2019*). Even after controlling for Axis I comorbidities, the avoidant, borderline, paranoid, schizoid, and schizotypal personality disorders (especially borderline personality disorder) all increased the risk of depression (*Skodol et al., 2011*). The cluster B personality disorders have been found to predict the severity and duration of depression, whereas the cluster C

personality disorders predict its chronicity (*Iacoviello et al., 2007*). In addition, personality disorders also have been shown to be associated with the effects of treatment for depression. For example, compared to patients with major depression, patients with both borderline personality disorder and major depression have poor self-regulation ability, which affects their treatment (*Kim et al., 2018*). A comorbidity of a personality disorder makes the treatment of major depression more complex, and it is associated with higher rates of recurrent depression and hospital readmission, compared to patients who only have major depression (*Wiegand & Godemann, 2017*). Personality disorders are also associated with significantly greater severity of self-harm, overall psychopathology, and impairment (*Ayodeji et al., 2015*). However, the mechanism underlying the relationship between a personality pathology and depression is unclear. In addition, a definitive diagnosis of a personality disorder requires long-term follow-up observations and multiple evaluations, so it is not easy to diagnosis personality disorders for inpatients with depression. Moreover, as inpatients with depression are mostly in the acute phase of the disease, it is impossible to exclude the influence of depression on screening for personality disorders. Therefore, we used the PDQ-4+ questionnaire to screen for personality pathology in this study.

Sleep difficulty is a crucial public health problem that involves daytime impairments and poor nighttime sleep. Insomnia is associated with brain function, cognitive and emotional effects, and physical health (*Ji et al., 2019*), and sleep loss can disrupt regions of the brain involved in affective regulation (*Kahn-Greene et al., 2007*). Recent studies have shown that insomnia not only reduces the quality of life, but is linked to the development of diseases (*Otaka et al., 2019*). Most mental disorders are associated with sleep continuity problems, abnormal depth of sleep and rapid-eye movement sleep may further play an important role in psychiatric comorbidity processes (*Baglioni et al., 2016*). Insomnia is closely related to depression (*Baglioni et al., 2011*). For example, a cohort study of elderly Asian persons found depression is associated with a number of sleep-related issues (*Yu et al., 2016*). This relationship was confirmed by a recent longitudinal study that showed poor sleep quality was significantly associated with greater symptoms of depression and anxiety, although the study involved two different populations (*Okun et al., 2018*). In addition, pathological personality traits are associated with insomnia (*Akram et al., 2018*; *Hafizi, 2013*). Features of cluster A and C personality disorders are linked to poorer sleep quality-related daytime functioning, fatigue, and estimated nightly wake-time (*Ruiter et al., 2012*). A cross-sectional study also reported that personality disorders, especially borderline personality disorder, were associated with insomnia (*Oltmanns, Weinstein & Oltmanns, 2014*). Moreover, the symptoms of borderline personality disorder interact with chronic sleep problems to predict elevated social/emotional, cognitive, and self-care deficits (*Selby, 2013*).

Previous studies have examined the relationship between personality disorders and depression—including research from the fields of epidemiology, epigenetics, and clinical treatments (*Gunderson et al., 2008*; *Newton-Howes et al., 2014*; *Weber, Giannakopoulos & Canuto, 2011*)—and their results have shown that personality disorders are associated with depression. Insomnia not only produces a variety of psychopathology symptoms

(*Kahn-Greene et al., 2007*), it is also closely related to the development of depression (*Baglioni et al., 2011*). However, the relationships among insomnia, specific personality pathology, and the severity of depression have not been directly studied in a single Chinese sample. Thus, this cross-sectional study aimed to compare differences in specific personality pathology and insomnia between individuals with depression and healthy controls, and to explore the relationships among specific personality pathology, insomnia, and depression severity. We hypothesized that specific personality pathology and insomnia would be associated with depression and that insomnia mediated the association between specific personality pathology and depression.

## MATERIALS AND METHODS

### Participants

The present study used a case-control design. Depressed patients between 18 and 60 years of age were consecutively recruited from the Inpatient Department of Mental Health of the Second Affiliated Hospital of Guangxi Medical University in Guangxi, China. A total of 69 inpatients with depression (mean age = 33.06 years, SD = 13.68) were recruited. The diagnosis of depression was made by professional psychiatrists using the Structured Clinical Interview based on the Diagnostic and Statistical Manual of Mental Disorders, Fourth Edition (SCID for DSM-IV), and inpatients who had a Hamilton Depression Rating Scale-24 (HAMD-24) score ≥ 20 and a Hamilton Anxiety Rating Scale (HAMA-14) score ≤ 7 were included. Any inpatients who met one of the following criteria were excluded: (1) they had a current infection, somatic trauma, or an autoimmune disease; (2) they had another psychiatric Axis-I disorders (except depression) after SCID screening; (3) they had a family history of a genetic disorder or bipolar disorder; (4) they had an organic brain disorder, mental retardation, or a major physical illness; or (5) they were pregnant or lactating. Inpatients recently had transfusion therapy were also excluded.

We also recruited 69 healthy controls (mean age = 35.23 years, SD = 12.18) from the same hospital through posters placed in the Medical Examination Center. The control group was selected using a 1:1 age matching ratio (±3 years). The inclusion criteria for the control group were having a Hamilton Depression Rating Scale (HAMD-24) score < 8 and a HAMA-14 score ≤ 7. People with a current or past mental illness, any major physical illness, genetic disease, or a family history of mental illness were excluded.

All the participants were given detailed information about the content and aims of the study, and the individuals then gave their written informed consent to participate in the study. The study was approved by the ethics committee of Guangxi Medical University (Approval number: 20160302-13).

### Measures

#### Insomnia

The Athens Insomnia Scale (AIS), which is a psychometric instrument based on the Tenth Revision of the International Classification Diseases (ICD-10), was used to assess the severity of insomnia during the past month (*Chung, Kan & Yeung, 2011*). This is an 8-item self-report scale. Item examples include: "Sleep induction (time it takes you to fall asleep

after turning-off the lights)" and "Functioning (physical and mental) during the day." Participants rated their insomnia on a scale from 0 to 3, with 0 corresponding to "no problem at all" and 3 corresponding to "very serious problem." The total score ranged from 0 to 24, with higher scores indicating more severe insomnia (*Soldatos, Dikeos & Paparrigopoulos, 2000*). A score of 6 or higher suggests participants have insomnia (*Soldatos, Dikeos & Paparrigopoulos, 2003*). The AIS is a reliable and valid tool, which has been widely used to test insomnia in different languages. The Cronbach's alpha of the AIS was 0.81 in a previous Chinese study (*Chung, Kan & Yeung, 2011*), and the Cronbach's alpha of the AIS was 0.932 in our study.

### Personality pathology

The Personality Diagnostic Questionnaire (PDQ-4+) (*Hyler et al., 1992*) was used to evaluate symptoms of pathological personality, based on DSM-IV criteria. This questionnaire screens 12 types of personality disorders (i.e., paranoid, schizoid, schizotypal, histrionic, narcissistic, borderline, antisocial, avoidant, dependent, obsessive, depressive, and negativistic personalities). The PDQ-4+ is a self-report questionnaire that consists of 107 true-false items; for instance, "Some people think that I take advantage of others" and "I have accomplished far more than others give me credit for." Higher subscale scores indicate a greater likelihood of having a certain type of personality disorder. This tool is a valid and reliable screening instrument for personality disorders (*Yang et al., 2000*). The PDQ-4+ Cronbach's alpha was 0.944 in the current study.

### Depression severity

The Hamilton Depression Rating Scale (HAMD) (*Hamilton, 1960*) was used by trained clinical psychologists to assess the severity of the participants' depression during the past 2 weeks. The scale contains 24 questions, with each question having five responses options, which are rated from 0 to 5; higher scores indicate greater severity of depression. This scale is known to have satisfactory reliability and validity to assess depression (*Huang et al., 2012*).

### Anxiety severity

Trained clinical psychologists used the 14-item Hamilton Anxiety Scale (HAMA-14) to evaluate the severity of anxiety of all participants during the current or most recent week. This scale, which has 14 items, has good psychometric properties, and includes psychic anxiety and somatic anxiety domains (*Watson, Clark & Tellegen, 1988*). Participants rate each item on a scale from 0 to 4, with higher scores indicating greater severity of anxiety. A total score higher than 7 indicates a person probably has anxiety; therefore, a score ≤ 7 was one of the inclusion criteria for the entire sample in our study.

   All participants were personally interviewed by clinical psychologists from the Mental Health of the Second Affiliated Hospital of Guangxi Medical University in Guangxi, who were trained in a standard way to perform face-to-face interview. All participants were interviewed upon their consent, and those who could not read the questionnaire were assisted by the interviewer.

## Statistical analysis

SPSS version 23.0 was used to enter and analyze the data. The chi-square test and Fisher's exact test were used to compare differences in the frequency of categorical variables (e.g., gender, ethnicity, and marital status) between the depressed and control groups. Student's $t$-test was used to analyze differences in insomnia (AIS scores), severity of depression (HAMD-24 scores) and each personality pathology (determined by PDQ-4+ subscale scores) between the two groups, and Spearman's rank correlation was used to measure the associations among the AIS, HAMD-24, and the scores for each type of personality pathology. Multivariate linear regression was used to analyze the degree to which insomnia and each type of personality pathology predicted the severity of depression severity, controlling for age, gender, ethnicity, education, income, and marital status. Finally, we conducted mediation analysis using AMOS 23.0, with HAMD-24 scores as the dependent variable, and AIS scores as the mediating variable; the analysis met the four criteria of Baron and Kenny's method (*Baron & Kenny, 1986*). Bootstrapping was used to test if AIS scores mediated the association between specific personality pathology scores and HAMD-24 scores. A $p < 0.05$ was considered to be statistically significant in all the analyses.

## RESULTS

### Sociodemographic characteristics

Table 1 shows the main sociodemographic characteristics of the depression and control groups. The patients with depression and the healthy controls did not differ with respect to age, gender, ethnic group, marital status, education, or income ($p > 0.05$).

HAMD-24 scores ($\chi^2 = -40.347$, $p < 0.001$) and AIS scores ($\chi^2 = -11.368$, $p < 0.001$) were significantly higher in the depression group than in the healthy control group. Significant group differences were found for all types of personality pathology scores, except antisocial personality: Compared to the healthy controls, inpatients had higher personality pathology scores ($p < 0.001$) (see Tables 1 and 2).

### Associations between each type of personality pathology, insomnia, and depression

In the entire sample, AIS scores had a significant positive correlation with HAMD-24 scores ($r = 0.620$, $p < 0.001$). With the exception of antisocial personality, all the other personality pathology scores were positively correlated with AIS scores and HAMD-24 scores (see Table 3). These linear correlations did not exist in the depression group or in the control group.

### Regression analysis for predicting depression severity

The multivariate regression that was conducted to test the ability of insomnia (AIS scores) and each type of personality pathology (PDQ-4+ subscale scores) to predict depression severity (HAMD-24 scores) found insomnia, borderline personality, and passive-aggressive personality significantly predicted the severity of depression. Table 4 shows, after adjusting for covariates, that the AIS scores ($\beta = 0.439$, $p < 0.001$), borderline

**Table 1 Sociodemographic characteristics, AIS scores, and HAMD-24 scores of all participants (N = 138): patients with depression vs. controls.**

| Variable | Depression | | Controls | | $\chi^2/t$ | p |
|---|---|---|---|---|---|---|
| | n = 69 | | n = 69 | | | |
| | n | % | n | % | | |
| Age (M ± SD) | 33.06 ± 13.68 | | 35.23 ± 12.18 | | 0.986 | 0.326 |
| Gender | | | | | | |
| Male | 29 | 42.0 | 35 | 50.7 | 1.049 | 0.306 |
| Female | 40 | 58.0 | 34 | 49.3 | | |
| Ethnic group | | | | | | |
| Han | 39 | 56.5 | 33 | 47.8 | 1.045 | 0.307 |
| Zhuang | 30 | 43.5 | 36 | 52.2 | | |
| Marital status | | | | | | |
| Single | 34 | 49.3 | 26 | 37.7 | – | 0.418 |
| Married | 32 | 46.4 | 40 | 58.0 | | |
| Divorced or widowed | 3 | 4.3 | 3 | 4.3 | | |
| Income per person (yuan/month) | | | | | | |
| 0–999 | 18 | 26.1 | 28 | 40.6 | 3.724 | 0.293 |
| 1,000–1,999 | 14 | 20.3 | 10 | 14.5 | | |
| 2,000–2,999 | 12 | 17.4 | 8 | 11.6 | | |
| ≥3,000 | 25 | 36.2 | 23 | 33.3 | | |
| Educational status | | | | | | |
| <6 years | 5 | 7.2 | 10 | 14.5 | 2.560 | 0.278 |
| 6–9 years | 33 | 47.8 | 26 | 37.7 | | |
| ≥9 years | 31 | 44.9 | 33 | 47.8 | | |
| AIS scores (M ± SD) | 14.14 ± 5.50 | | 4.67 ± 4.21 | | −11.368 | <0.001 |
| HAMD-24 scores (M ± SD) | 27.32 ± 4.79 | | 3.48 ± 1.08 | | −40.347 | <0.001 |

Note:
AIS, Athens Insomnia Scale; HAMD-24, 24-item Hamilton Depression Scale.

personality scores (β = 1.512, p = 0.003), and passive-aggressive personality scores (β = 1.538, p = 0.003) significantly predicted HAMD-24 scores.

## Mediating effect of insomnia on the association between personality pathologies and depression

We conducted a series of mediation analyses based on the results of correlation analyses and multivariate regression analysis. We found that insomnia had a significant partial mediating effect on the relationship of depression severity with borderline personality and passive-aggressive personality, based on 5,000 bootstrap samples. The standardized direct effects of borderline personality and passive-aggressive personality on depression severity were 0.553 and 0.548, respectively; the standardized indirect effects of borderline personality and passive-aggressive personality on depression severity through insomnia

**Table 2 Comparison of the number of personality pathologies between depressive inpatients and controls.**

| Numbers of personality pathologies | Depression (n = 69) | | Controls (n = 69) | | t | P |
|---|---|---|---|---|---|---|
| | Mean | SD | Mean | SD | | |
| PDQ-4+ scores | 51.09 | 18.29 | 18.25 | 7.37 | −13.833 | <0.001 |
| Paranoid PP | 3.29 | 2.00 | 1.26 | 1.35 | −6.988 | <0.001 |
| Schizoid PP | 3.46 | 2.08 | 0.75 | 0.63 | −10.346 | <0.001 |
| Schizotypal PP | 4.36 | 2.25 | 1.41 | 0.99 | −9.995 | <0.001 |
| Antisocial PP | 1.80 | 1.82 | 1.39 | 1.32 | −1.500 | 0.136 |
| Borderline PP | 5.83 | 2.49 | 1.33 | 1.30 | −13.277 | <0.001 |
| Histrionic PP | 4.06 | 1.81 | 2.06 | 1.51 | −7.032 | <0.001 |
| Narcissistic PP | 4.33 | 2.18 | 1.91 | 1.28 | −7.951 | <0.001 |
| Avoidant PP | 5.41 | 2.37 | 2.07 | 1.08 | −10.654 | <0.001 |
| Dependent PP | 4.13 | 2.29 | 1.04 | 1.30 | −9.746 | <0.001 |
| Obsessive-Compulsive PP | 6.36 | 2.45 | 1.99 | 1.75 | −12.070 | <0.001 |
| Depressive PP | 3.22 | 1.70 | 1.64 | 0.94 | −6.767 | <0.001 |
| Passive-Aggressive PP | 4.84 | 1.89 | 1.39 | 1.18 | −12.857 | <0.001 |

Note:
PDQ-4+, Personality Diagnostic Questionnaire; PP, Personality pathology.

were 0.205 and 0.208, respectively (see Table 5). Figures 1 and 2 show the path coefficients calculated by AMOS.

# DISCUSSION

The main aim of this study was to explore the associations among each type of personality pathology, insomnia, and the severity of depression. Personality traits may develop into personality disorders under certain circumstances and conditions, leading to significantly decreased adaptation, affect, and social and professional functioning, and may even cause clinical problems (*Association, 2000*). With the exception of antisocial personality, all the personality pathology scores in our study were significantly higher in the depression group than in the healthy control group, which shows that personality pathologies are common among depressed inpatients. This result is partially in line with a study by *Kounou et al. (2013)* that found patients in Togo who were treated for major depression had higher scores for all types of personality disorders and had more personality disorders than the controls did. Furthermore, several other studies have detected generally high rates of personality disorders in patients with depression. For example, *Sanderson et al. (1992)*, which used the Axis I SCID-P and the Axis II SCID-II to assess the personality disorders of patients with major depression, found 50% of these patients had at least one personality disorder. A Thai study similarly found that 77% of depressed patients suffered from at least one personality disorder and 60% had two or more personality disorders (*Wongpakaran et al., 2015*).

The present study showed that in the entire sample, all types of personality pathologies but antisocial personality were positively correlated with depression severity, which is consistent with the findings of some prior studies. Personality pathology is frequently

**Table 3 Correlations between the main study variables ($N = 138$).**

| | Paranoid PP | Schizoid PP | Schizotypal PP | Antisocial PP | Borderline PP | Histrionic PP | Narcissistic PP | Avoidant PP | Dependent PP | Obsessive-Compulsive PP | Depressive PP | Passive-Aggressive PP | AIS scores | HAMD-24 scores |
|---|---|---|---|---|---|---|---|---|---|---|---|---|---|---|
| Paranoid PP | 1 | | | | | | | | | | | | | |
| Schizoid PP | 0.584** | 1 | | | | | | | | | | | | |
| Schizotypal PP | 0.736** | 0.674** | 1 | | | | | | | | | | | |
| Antisocial PP | 0.594** | 0.139 | 0.322** | 1 | | | | | | | | | | |
| Borderline PP | 0.741** | 0.640** | 0.716** | 0.367** | 1 | | | | | | | | | |
| Histrionic PP | 0.633** | 0.394** | 0.582** | 0.514** | 0.619** | 1 | | | | | | | | |
| Narcissistic PP | 0.533** | 0.542** | 0.689** | 0.271** | 0.629** | 0.530** | 1 | | | | | | | |
| Avoidant PP | 0.629** | 0.613** | 0.642** | 0.310** | 0.772** | 0.553** | 0.527** | 1 | | | | | | |
| Dependent PP | 0.515** | 0.575** | 0.469** | 0.224** | 0.650** | 0.509** | 0.606** | 0.601** | 1 | | | | | |
| Obsessive-Compulsive PP | 0.613** | 0.577** | 0.625** | 0.239** | 0.728** | 0.598** | 0.609** | 0.652** | 0.710** | 1 | | | | |
| Depressive PP | 0.549** | 0.433** | 0.520** | 0.453** | 0.634** | 0.563** | 0.448** | 0.618** | 0.495** | 0.566** | 1 | | | |
| Passive-aggressive PP | 0.567** | 0.749** | 0.634** | 0.223** | 0.655** | 0.497** | 0.591** | 0.656** | 0.710** | 0.605** | 0.397** | 1 | | |
| AIS scores | 0.447** | 0.585** | 0.511** | 0.101 | 0.551** | 0.478** | 0.397** | 0.459** | 0.505** | 0.552** | 0.367** | 0.612** | 1 | |
| HAMD-24 scores | 0.490** | 0.694** | 0.609** | 0.080 | 0.687** | 0.480** | 0.570** | 0.624** | 0.592** | 0.614** | 0.420** | 0.701** | 0.620** | 1 |

**Notes:**
** $P < 0.01$.
AIS, Athens Insomnia Scale; HAMD-24, 24-item Hamilton Depression Scale; PP, Personality pathology.
**Table 4 Multivariate regression results for the predictors of depression severity (N = 138).**

| Independent variables | Dependent variable | Unadjusted β | Adjusted β[#] | SD | t | p |
|---|---|---|---|---|---|---|
| AIS scores | Depression | 0.412 | 0.439 | 0.112 | 3.611 | <0.001 |
| Borderline PP | Depression | 1.505 | 1.512 | 0.502 | 3.010 | 0.003 |
| Passive-Aggressive PP | Depression | 1.455 | 1.538 | 0.499 | 3.080 | 0.003 |

Notes:
[#] Adjusted for age, gender, education, income, ethnicity, and marital status.
AIS, Athens Insomnia Scale; PP, Personality pathology.

**Table 5 Results of the mediation analysis for the predictors of depression severity (N = 138).**

| Independent variables | Mediator variable | Direct effect | | | Bootstrapping bias-corrected 95% CI | | Indirect effect | | | Bootstrapping bias-corrected 95% CI | |
|---|---|---|---|---|---|---|---|---|---|---|---|
| | | Effect | SE | Z | Lower | Upper | Effect | SE | Z | Lower | Upper |
| Borderline PP | Insomnia | 0.553 | 0.059 | 9.373 | 0.435 | 0.664 | 0.205 | 0.039 | 5.256 | 0.136 | 0.292 |
| Passive-Aggressive PP | | 0.548 | 0.069 | 7.942 | 0.412 | 0.684 | 0.208 | 0.045 | 4.622 | 0.125 | 0.303 |

Note:
PP, Personality pathology.

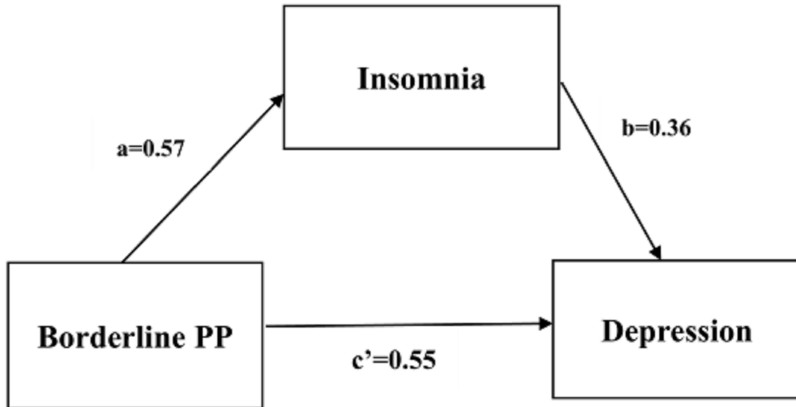

**Figure 1 Partial mediating relation of insomnia between borderline PP and depression.**

associated with internalized and externalized symptoms, social relationship problems, impulsive behavior, and comorbidities (*Sperandeo et al., 2019*). Longitudinal cohort studies have reported that the presence of personality disorders predicts the occurrence of later depression (*Moran et al., 2016*) and relapses of depression (*Alnaes & Torgersen, 1997*). Another study also showed that personality disorders, such as the avoidant, borderline, histrionic, and schizotypal types, can enhance the persistence of major depressive disorder (*Skodol et al., 2011*). In addition, our study further showed that borderline personality and passive-aggressive personality were the strongest positive predictors of depression severity. The possible reasons for this were: first, decreased positive affect (e.g., feeling enthusiastic and excited) and increased negative affect (e.g., feeling nervous and upset) are important dimensions and core features of major
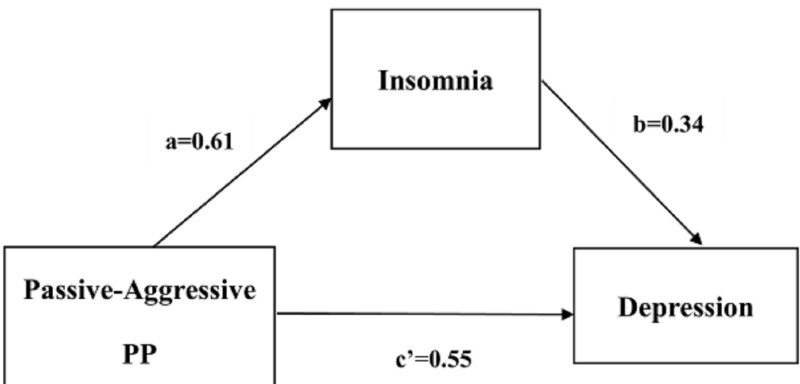

**Figure 2 Partial mediating relation of insomnia between passive-aggressive PP and depression.**

depressive disorder (*Watson, Clark & Tellegen, 1988*). Several personality pathologies, including the traits of borderline and passive-aggressive personality disorders have been found to have very reliable associations with negative mood (*Farmer, Nash & Dance, 2004*). Second, the dysfunctional beliefs, cognitive reactivity, and rumination of patients with major depression are associated with personality pathologies (*Van Rijsbergen et al., 2015*). Research has shown that the traits exhibited in borderline personality disorder are significantly related to both sociotropy and autonomy, and the traits exhibited in passive-aggressive personality disorder are specifically related to autonomy (*Morse, Robins & Gittes-Fox, 2002*). Sociotropy and autonomy are sets of beliefs, concerns, and behavioral tendencies that have been reported to increase vulnerability to depression and other psychopathologies and can affect responses to treatment (*Beck, 1983*; *Cappeliez, 1993*). Our data also highlight: (1) that patients with depression were more likely to have insomnia than healthy controls were; and (2) that the severity of insomnia was associated with depression severity, in the entire sample. Research has shown that insomnia symptoms (e.g., difficulties with sleep onset and/or sleep maintenance) cause disturbances in circadian rhythms (*Zee, Attarian & Videnovic, 2013*), which increase depressed mood (*Booker et al., 1991*). Furthermore, the presence of depressed mood is associated with abnormal circadian activity in the hypothalamic-pituitary-adrenal (HPA) axis (*Dowlati et al., 2010*). Our findings are consistent with many studies that have found insomnia is associated with depression and that it appears to be a risk factor for depression (*Fernandez-Mendoza et al., 2015*; *Okun et al., 2018*). A study by *Fernandez-Mendoza et al. (2015)* which used psychometric and polysomnographic data to measure the sleep quality of a sample of the general population, found that persistence and worsening poor sleep, or insomnia, were significant predictors of the incidence of depression. Therefore, our findings provide further evidence confirming that certain personality pathologies (i.e., borderline and passive-aggressive personality traits) and insomnia are risk factors for depression.

Interestingly, the results of the mediation analysis showed that insomnia partially mediated the relationships of borderline personality and passive-aggressive personality

with the severity of depression, which may contribute to our understanding of the pathways from personality disorders to depression. These results suggest that borderline personality and passive-aggressive personality may promote insomnia, and this indirectly increases the negative influence of personality disorders on depression. This may be partly explained by a biologically plausible mechanism—the dopamine system. Several studies have shown that dopamine system dysfunction is associated with the symptoms and treatment of depression (*Dailly et al., 2004*). For example, anhedonia, which is a common symptom of depression that is associated with the dysfunction of dopamine, is thought to result from reduced $D_2/D_3$ receptor binding in the nucleus accumbens (*Papp, Klimek & Willner, 1994*). Some studies have also shown that dopamine dysfunction is related to emotional dysregulation, impulsivity, and cognitive-perceptual impairments in people with borderline personality disorder (*Friedel, 2004*). Moreover, borderline personality traits, such as impulsivity, affective instability, anger, and stress-related dissociation are all related to a person's temperament and uniquely associated with sleep disturbances (*Oltmanns, Weinstein & Oltmanns, 2014*). At the same time, insomnia has been demonstrated to be associated with dopamine dysfunction (*Finan & Smith, 2013*). Insomnia causes abnormalities in amygdala-frontal functional connectivity, which predict the development of internalized psychopathologies (*Klumpp, Hosseini & Phan, 2018*) and have a negative effect on mood and cognitive function, and increase the risk of mortality (*Roth, 2007*). At present, although no research has indicated there is a direct relationship between the passive-aggressive personality and insomnia, the DSM-IV description of passive-aggressive personality disorder emphasizes the pattern of sullen and irritable moods and negative attitudes (*GUZE & Samuel, 1995*). A person who tends to internalize emotions may develop chronic emotional arousal that develops into physiological hyperarousal, and then insomnia (*Kales et al., 1976*). And poor sleep is known to be a risk factor for the development and maintenance of mood disorders (*Gillin, 1998*). Personality disorders are aggravated by poor sleep and lead to higher levels of functional impairment (e.g., social/emotional, cognitive, and self-care impairments) (*Selby, 2013*) that may contribute to depression. In contrast, a prospective study indicated that the effect of cluster A and B personality disorders on subsequent depression were mediated by stress; that is, cluster A and B personality disorders were associated with increased episodic stress and chronic interpersonal stress, which in turn, were linked to subsequent depression (*Daley et al., 1998*). Taken together, our study may provide another way for researchers to understand the pathology from personality disorders to depression. Currently, however, it is unclear what mechanisms underlie the mediating role of insomnia on the relationship between personality pathology and depression. Future research is needed to further elucidate the relationship and mechanism.

According to our results and the previous evidence we reviewed, the presence of a personality disorder is not only associated with depression severity, but the comorbidity of a personality disorder can complicate treatment and worsen the prognosis of depression (*Wiegand & Godemann, 2017*; *Zimmerman, Rothschild & Chelminski, 2005*) and other unhealthy outcomes, including self-harming behaviors (*Ayodeji et al., 2015*). Therefore, understanding the relationship between personality disorders and depression

has clinical importance for prognostic accuracy and individualized interventions for depression. Furthermore, our findings show that insomnia not only predicts depression severity, but also partially mediates the relationship between specific pathological personality traits and depression severity, which indicates that we could slow the development of depression and relieve depression severity through improving sleep quality, as well as helping us to understand the pathways from personality disorders to psychopathology.

Like all studies, our study has some limitations. First, the cross-sectional design of our study is a limitation because it makes it impossible to determine the causal relationships among insomnia, personality pathology, and depression severity. Although the research evidence supporting our model was discussed in the introduction, the partial mediating effect of insomnia on the association between personality pathology and depression is a hypothesis that needs to be more thoroughly explored in prospective and longitudinal research in future. Second, it is common for a diagnosis of a personality disorder to co-occur with major depressive disorder (*Zheng et al., 2019*). However, our inpatients with depression were not examined by a clinician for the diagnosis of a comorbid personality disorder. Therefore, in addition to being affected by the selection bias of the case-control study, these findings were also affected by the confounding bias of this design. A prospective cohort study would have allowed us to clarify the relationships among personality disorders, insomnia, and depression. Moreover, the assessment of pathological personality in the entire sample was conducted by screening participants with the PDQ-4+ scale, rather than through a clinical evaluation and diagnosis by psychologists using a structured interview; so the PDQ-4+ scores merely indicate the likelihood of having personality disorders. Third, using a self-report scale to assess insomnia may involve a certain degree of recall bias. To evaluate insomnia more precisely, a future study using some objective sleep assessments (e.g., polysomnography) is required. Fourth, these findings may not be representative, because the sample was only from China. Despite these limitations, our study establishes links among personality pathology, insomnia, and depression.

## CONCLUSIONS

These findings further demonstrate that borderline and passive-aggressive personality traits and insomnia are risk factors for depression severity, which should enable researchers to understand the risk factors of depression severity and clinicians to focus on evaluating specific personality disorders and sleep problems to provide optimal interventions and treatments to relieve the severity of depression in a timely way. In addition, to the best of our knowledge, this is the first study to examine the mediating role of insomnia on the relationships between personality disorders and depression in a Chinese sample, which may provide researchers with a way to understand the pathways from personality pathology to psychopathology. However, because our findings are based on cross-sectional data, causality remains unclear. Prospective studies are needed in the future to verify the mediating role of insomnia on the association between pathological personality and depression.

### Funding
This work was funded by the National Natural Science Foundation of China (No. 81660569) and the Natural Science Foundation of Guangxi Province (No. 2017GXNSFAA198212). The funders had no role in study design, data collection and analysis, decision to publish, or preparation of the manuscript.

### Grant Disclosures
The following grant information was disclosed by the authors:
The National Natural Science Foundation of China: 81660569.
The Natural Science Foundation of Guangxi Province: 2017GXNSFAA198212.

### Competing Interests
The authors declare that they have no competing interests.

### Author Contributions
- Fenglan Chen performed the experiments, analyzed the data, prepared figures and/or tables, and approved the final draft.
- Xiujin Lin conceived and designed the experiments, performed the experiments, authored or reviewed drafts of the paper, and approved the final draft.
- Yuli Pan performed the experiments, prepared figures and/or tables, and approved the final draft.
- Xuan Zeng conceived and designed the experiments, prepared figures and/or tables, and approved the final draft.
- Shengjie Zhang performed the experiments, prepared figures and/or tables, and approved the final draft.
- Hong Hu analyzed the data, prepared figures and/or tables, and approved the final draft.
- Miaoyu Yu performed the experiments, authored or reviewed drafts of the paper, and approved the final draft.
- Junduan Wu conceived and designed the experiments, authored or reviewed drafts of the paper, and approved the final draft.

### Human Ethics
The following information was supplied relating to ethical approval (i.e., approving body and any reference numbers):
The study was approved by the ethics committee of Guangxi Medical University (Approval number: 20160302-13).

### Data Availability
Raw measurements are available as a Supplemental File.

## Supplemental Information

Supplemental information for this article can be found online at http://dx.doi.org/10.7717/peerj.11061#supplemental-information.

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
