# Peer review of "Insomnia partially mediates the relationship between pathological personality traits and depression: a case-control study"

_PeerJ, doi:10.7717/peerj.11061_

## Round 0.1 · original submission · Major Revisions

Dear Dr. Chen,

I look forward to your resubmission addressing the issues discussed by our reviewers.

Reviewer 1 ·

Basic reporting

Of an extremely high standard of writing with no English language errors.

Experimental design

Just one small thing to clarify: what is the "positive threshold" of the HAMA-24 scale and HAMD-24 scale as referenced in line 126? Is there a numerical value- if so, what is it for each scale?

Validity of the findings

It would be important for the authors to emphasise in the abstract and conclusions that their findings of linear correlations between AIS, PDQ-4 and HAMD scales are in the total cohort (depressed and non-depressed), not just depressed patients.

It might also be worthwhile to present or mention data about whether these linear correlations also exist within each of the two groups (e.g. within the depressed patients, and within the healthy controls).

Finally it is important to for the authors to mention in their title, abstract and conclusions that their findings are of partial mediation, not complete mediation, of insomnia on the link between personality disorders and depression. After all, the correlation between PDQ-4 and HAMD-scales remained statistically significant after the mediation analysis was performed, albeit just of a lower magnitude.

Reviewer 2 ·

Basic reporting

- generally well-written in professional English
- statistical indices (e.g. r) are missing at some points
- a high number of Chinese studies is cited, non-Chinese studies should also (or preferentially) by included
- differentiation between diagnosis, syndrome and symptoms remains unclear (confounded)

Experimental design

- Case control-study → possible influence on interpretation of results should be discussed more clearly
- If only people with depression were examined, not with comorbid (diagnosed) PD; the resulting selection bias should be clearly stated and discussed
- Exclusion criteria should be stated more clearly (only insomnia and PD symptoms?)
- The assessment methods should be described in more detail (reliability/validity: two self-report questionnaires and depression scale; assessed by trainer/qualification?)
- Heterogeneity and broad variety of PD symptoms make interpretation difficult
- Causality remains unclear due to study design (partly mentioned by authors)
- statistical effects of mediation should be cross-checked (e.g. subdividing samples vs. jackknifing); furthermore the model is not unique (comparison vs. no mediation, other mediation, interaction, and non-linear mediation should at least be discussed)
- bidirectional relationships should be shown and reported as what they are

Validity of the findings

- relationships as found in the study are plausible if not trivial
- limitation of interpretation of results (validation necessary)
- limited generalization
- importance and clinical relevance questionable

Additional comments

- The manuscript is generally well-written, the topics of PD and sleep quality are still timely.
- Assessment methods are weak and require more justification (reliability, validity, time-economy).
- Statistical model is plausible and preferred but not unique (e.g., a mediation of the relationship of insomnia and depression by PD symptoms).
- Focussing on Chinese literature could either be a source of bias or a limitation (generalizability) at the present time.
- All areas of interest (i.e. PD, insomnia, depression) are much more specifically elaborated than assessed and dealed with here (e.g., recent discussion ref. to ICD-11/DSM-5, biological models; chronobiological apporaches, subtypes/cognitive and somatic dimensions of depression etc.).
- Thus, implications of the present study are questionable (for GPs? clinicians? researchers?).
- What does the study add to our knowledge or clinical practices?.

Reviewer 3 ·

Basic reporting

The extensive use of abbreviations from start to finish makes the paper hard to follow. This includes the tables, which are messy and somewhat difficult to understand. The references are mostly satisfying, but the overall context and a thorough background is lacking. In the first paragraph of the introduction, it is stated that "Depression is a common and recurrent, but often neglected mental disorder." Depression is one of the most researched and well-known mental disorders in the world. It is not neglected in neither research or in public health care. This claim is almost directly false, which literature supports this statement?

Experimental design

The overall aim of the paper is interesting, but it lacks a clear rational. Which symptoms personality disorder(s) do the authors refer to? This is a concept with an immense span ranging from antisocial PD to borderline and avoidant PD. Some of them will be stronger linked to both insomnia and depression than others, but this is not communicated at all.

The use of case-control is a major strenght of the paper, but the cross-sectional design makes it very difficult (if not impossible) to draw conclusions about mediating effects. This study would have benefitted from a longitudinal design with baseline and follow-up measurement to establish the link between PD and depression through insomnia. Lastly, the method section should provide examples of items/questions to each instrument used in the study. This would have given the reader more information about which symptoms of PD that actually were measured.

Validity of the findings

Due to the cross-sectional design, the study do not provide information about true mediating effects. The conclusion is vague and do not really provide any new insight on the relation between PD symptoms (which symptoms?), insomnia and depression.

Additional comments

The study would have greatly benefitted from a longitudinal design. Please consider adding a follow-up measurement of the constructs to strenghten the conclusion.

---

## Round 0.2 · accepted · Accept

Thank you very much for the detailed and careful revision - your paper is now ready to see the light of day.

Reviewer 1 ·

Basic reporting

No further concerns

Experimental design

No further concerns

Validity of the findings

No further concerns